# Study on the Mechanism of Elastic Instability Caused by Natural Growth in Orthotropic Material

**DOI:** 10.3390/ma15207059

**Published:** 2022-10-11

**Authors:** Diquan Wu, Liwen He

**Affiliations:** Key Laboratory of Impact and Safe Engineering, Ministry of Education, School of Mechanical Engineering and Mechanics, Ningbo University, Ningbo 315211, China

**Keywords:** non-uniform in-plane growth, morphological evolution, elastic instability, organizational form, finite element analysis

## Abstract

Compared to synthetic materials, naturally grown biological materials have more specific behavioral patterns and life connotations in their morphological evolution over millions of years of environmental evolution on Earth. In this paper, we investigate the physical mechanisms and manifestations of out-of-plane deformation instability. Firstly, the origin of the instability phenomenon caused by the growth of the leaf is introduced. Leaf instability problems are modeled using rectangular thin plates. Secondly, the variation in the critical intrinsic strain with the principal shear modulus is obtained by numerical solution. The post-buckling behavior of the growth instability is further analyzed by general static analysis, and we obtain the phase diagram of morphogenesis of thin plant organs as functions of the principal shear modulus and off-axis angle. The research results enhance the understanding of the mechanism of elastic instability caused by natural growth in orthotropic materials.

## 1. Introduction

Through billions of years of evolution, biological systems have developed a wide variety of morphology evolution schemes. Revealing these principles of morphological evolution not only elucidates the mechanisms underlying the evolution of living systems, but also lays the scientific foundation for bionics. Furthermore, we can develop advanced materials and systems by learning bionics from nature. There are many engineering systems based on plant inspiration. Examples include superhydrophobic surfaces designed with lotus leaf structures [1]; self-healing antifouling surfaces inspired by pigweed [2]; and drive systems inspired by the Venus flytrap [3,4,5]. The three-dimensional shapes of plant organs are extensive and diverse, and much work has been done to identify the genetic and chemical factors that achieve morphological evolution of plant organs by regulating cell division or cell expansion [6,7,8,9]. Although specific genes can control cell growth in a programmed manner, biomechanics also plays an important role in the formation of three-dimensional plant conformation.

Since the 18th century, there have been important contributions to the development of stability theory by famous scholars such as Liapunov, Timoshenko, Von-Kamen, and Xuesen Qian [10,11,12,13]. For some relatively simple one-dimensional rod structures, two-dimensional plane strain or plane stress structures, and axisymmetric structures, Euler stability theory and Cote stability theory can be applied [12], which can be solved by numerical methods. In addition to these methods, the phase field method and Monte Carlo method are also used for instability simulation of soft materials. He et al. [14] used the phase field method to simulate surface instability and interface detachment of thin films on soft substrates. Although existing computational methods are able to handle some critical buckling and initial post-buckling problems, the highly nonlinear problems during the transformation of soft materials to morphological instability still require more in-depth investigation. There is an urgent need to develop effective computational methods to deal with coupled mechanical–chemical–biological mechanisms.

In liquid crystal elastomers (LCEs), the combination of liquid crystallinity and rubber elasticity results in a strong coupling between macroscopic shape and molecular orientation [15]. In studies by Broer et al. [16,17] and White et al. [18], it was found that the inhomogeneous bending and torsional deformation of highly cross-linked rigid LC networks is thermally driven. They used a “hybrid” arrangement, where the nematic phase director has a continuous spatial gradient on the film, isotropic on one surface and planar on the other. In both structures, the gradient in the orientation causes inhomogeneous distortion, resulting in an initially flat sample that transforms into a three-dimensional structure. This suggests that LC networks have significant applications in thermally driven soft actuators [19].

Differential growth of plants has behavioral specificity and unique implications for life. Biological growth has many different physical mechanisms and manifestations at different spatial and temporal scales, such as phase changes, geometrical morphological transformations, etc. When the strain caused by growth reaches or exceeds a critical value, the leaf will transition from planar to curved equilibrium. Many scholars have studied the instability caused by the intrinsic strain of isotropic material leaves [20,21]. The study of instability due to non-uniform growth strain can provide insight into the morphogenesis of plant organs in nature. It will help inspire biomechanics, bionics, and flexible electronic components, among other fields.

In contrast, this paper is concerned with the instability of more general orthotropic material leaves under unevenly distributed intrinsic strains on the surface. The effect of the elastic principal axis of the material not coinciding with the growth direction is also considered. The rest of the paper is organized as follows. Section 2 presents the mechanical modeling of the leaf instability phenomenon, and the accuracy of the instability model is validated. In Section 3, the critical intrinsic strain and the corresponding critical instability mode of the leaf are obtained by finite element analysis. Furthermore, the linear combination of the first five orders of the critical instability mode is added as the initial geometric defect. We can obtain the phase diagrams of morphology evolution for different principal shear modulus and off-axis angle in orthotropic materials. In the last section, a brief summary of this paper is presented.

## 2. Finite Element Model

### 2.1. Geometric Model of Leaf Growth

In order to investigate the effect of an inhomogeneously distributed growth strain distribution on buckling, the leaf is modeled here using a rectangular thin plate. An *x*–*y* coordinate plane is established within the mid-plane, with the direction perpendicular to this surface as the *z* coordinate axis, as shown in Figure 1. *L* and *B* represent the length and width of the leaf.

Two different growth patterns exist in the general instability phenomenon caused by plant tissue growth and its associated applications. The first growth mode is spontaneous bending of the intrinsic curvature, which is also often used artificially by engineers and is characterized by direct bending intrinsic strain through the thickness direction. The second mode is an in-plane tensile growth phenomenon driven by tensile strain in the mid-plane film and all layers in the thickness direction. If the effect of only a single growth mode on the instability morphology is studied, it is referred to as a single growth mode. If two or more growth modes jointly affect the instability morphology, it is referred to as a multiple growth mode [22]. In this paper, only the second of these growth modes occurs in the leaf, and the distribution of the intrinsic strain on the surface within a given surface is considered.
(1)$$\varepsilon_{x}^{g}(y)=\beta {\left(\frac{y}{w}\right)}^{n}$$
Here, $\varepsilon_{x}^{g}(y)$ is the intrinsic strain field of the leaf within the surface along the length (*x*) direction. It is assumed that $\varepsilon_{x}^{g}(y)$ is distributed as a power function along the width (*y*) direction only. *w* is the distance from the mid-vein to the long edge of the leaf (*B* = 2*w*). The leaf edge (*y* = ±*w*) has the maximum intrinsic strain *β*. *n* is the surface power-law exponent of the intrinsic strain, which reflects the concentration of the distribution of the intrinsic strain along the width direction toward the long edge.

The total strain field of the model is obtained by overlaying the mid-surface strain field and the bending curvature field:(2)$$\{\varepsilon \}=\left\{\begin{array}{l} \varepsilon_{x}\\ \varepsilon_{y}\\ \varepsilon_{xy} \end{array}\right\}=\left\{\begin{array}{l} \varepsilon_{ox}\\ \varepsilon_{oy}\\ \varepsilon_{oxy} \end{array}\right\}+z\left\{\begin{array}{l} k_{x}\\ k_{y}\\ k_{xy} \end{array}\right\},$$
where {*ε_o_*} is the mid-plane strain field of the leaf and {*k*} is the bending curvature field of the leaf. Because the leaf intrinsic curvature component is assumed to be zero, the bending curvature is contributed by elastic bending.

Since the leaf cannot satisfy geometric compatibility of its surface in the in-plane intrinsic strain field only, the leaf induces internal elastic strain during growth. This makes the total strain field obtained by overlaying elastic strain with the intrinsic strain field satisfy the geometric compatibility equation [20]. In this paper, the possible plastic deformation of the material due to the intrinsic strain is always neglected during the growth of the plant elastic tissue, and the intrinsic linear relationship is considered to always hold. We treat the total strain field as a linear superposition of the elastic strain field and the intrinsic strain field only:(3)$$\left\{\begin{array}{l} \varepsilon_{x}\\ \varepsilon_{y}\\ \varepsilon_{xy} \end{array}\right\}=\left\{\begin{array}{l} \varepsilon_{x}^{e}\\ \varepsilon_{y}^{e}\\ \varepsilon_{xy}^{e} \end{array}\right\}+\left\{\begin{array}{l} \varepsilon_{x}^{g}\\ \varepsilon_{y}^{g}\\ \varepsilon_{xy}^{g} \end{array}\right\},$$
where {*ε^e^*} and {*ε^g^*} are the elastic strain field and the intrinsic strain field inside the leaf, respectively.

### 2.2. Constitutive Relation

Compared with isotropic materials, many plant tissues and organs in nature exhibit mechanical properties that are orthotropic. Therefore, the leaf tissues established in this section exhibit orthotropic material properties. In order to be closer to the general situation of natural growth, in this section, the constitutive relation of orthotropic materials is established, and the effect of off-axis angle on instability is considered. As shown in Figure 2, an element body is selected from the mid-plane of the leaf. The principal elastic axes of its orthotropic material are denoted axis 1 and axis 2, and the angle between one of the principal elastic axes and the *x*-direction is defined as the off-axis angle *θ*.

Based on the linear–elastic assumption, the material constitutive relationship is:(4)$$\left\{\begin{array}{l} \varepsilon_{x}^{e}\\ \varepsilon_{y}^{e}\\ 2\varepsilon_{xy}^{e} \end{array}\right\}=\left\{\begin{array}{l} S_{xx}\;S_{xy}\;S_{xz}\\ S_{yx}\;S_{yy}\;S_{yz}\\ S_{zx}\;S_{zy}\;S_{zz} \end{array}\right\}\cdot \left\{\begin{array}{l} \sigma_{x}\\ \sigma_{y}\\ \tau_{xy} \end{array}\right\}$$
(5)$$\left\{\begin{array}{l} \sigma_{x}\\ \sigma_{y}\\ \tau_{xy} \end{array}\right\}=\left\{\begin{array}{l} C_{xx}\;C_{xy}\;C_{xz}\\ C_{yx}\;C_{yy}\;C_{yz}\\ C_{zx}\;C_{zy}\;C_{zz} \end{array}\right\}\cdot \left\{\begin{array}{l} \varepsilon_{x}^{e}\\ \varepsilon_{y}^{e}\\ 2\varepsilon_{xy}^{e} \end{array}\right\}$$
where *σ_x_*, *σ_y_*, and *τ_xy_* are the three components of the stresses in the plate due to the elastic strain field. For the sake of brevity, the specific components of the flexibility {*S*} and stiffness {*C*} matrices appearing in this equation are given in the Appendix A of this paper.

In contrast to isotropic materials, we need to consider the effects of the orthotropic elastic off-axis angle and the material’s principal shear modulus on the unstable morphology at the same time.

### 2.3. Model Validation

Huang et al.’s [21] research on the post-buckling morphology of isotropic material leaves showed that for a parallel growth strand distribution with *n* = 2, when *β* = 0.05, the leaf began to exhibit a Twisting form until *β* = 0.15; then, it transitioned from a Twisting form to a Helical Twisting form. Their obtained values of intrinsic strain *β* taken at [0.02, 0.2] with the twist angle per unit length along the stem *αw* approximately satisfied:(6)$$\beta =\sqrt{1+{\left(\alpha w\right)}^{2}}-1$$

In Equation (6), Huang et al. [21] predicted that the formation of Twisting with *n* = 2 is positively favorable at lower intrinsic strain (*β* ≤ 0.2). The evolution of the post-buckling morphology of the leaf with intrinsic strain was demonstrated by finite element results, verifying the predictions of Huang et al. The significance of this study is to reveal the effect of the power-law exponent on the leaf buckling morphology. Careful energy analysis showed that at lower intrinsic strain (*β* ≈ 0.05), two local energy minimum states appear for both the Twisting and Saddle Bending configurations, and the strain energy of the Saddle Bending configuration is lower relative to that of the Twisting. Since buckling always selects the equilibrium path with lower strain energy, it was demonstrated in Huang’s report that Saddle Bending occurs preferentially to Twisting in the post-buckling form of the leaf (*n* = 2).

To ensure the accuracy of the finite element simulation results, a comparative study was first done. Then, further parametric study of the model in this paper was performed.

The numerical results of this paper are compared with those from Huang et al. [21] in Figure 3. It can be seen that the results in this paper are the same as Huang’s results, which verifies the correctness of the model and the research method derived in this paper.

As shown in Figure 4, the solution of this paper was compared with the variation pattern of the characteristic parameters of the post-buckling morphology obtained by Huang et al. [21]. The results are the same as Equation (6).

In the next section, the established orthotropic leaf growth instability problem is analyzed by finite element simulation.

## 3. Results and Discussion

### 3.1. Effect of Principal Shear Modulus on Critical Instability Conditions

In this section, an instability analysis is performed on a leaf with more general material properties of orthotropic material, and the effect of its in-plane principal shear modulus *G* on the critical instability conditions is obtained.

In this article, the morphogenesis of growing leaves was simulated using the commercial software ABAQUS. We utilized a thermal strain field to equivalently replace the intrinsic strain field, where a uniform non-zero thermal expansion coefficient was specified along the longitudinal direction and a non-uniform temperature field was specified as the resolved field. The leaf was modeled as an elastic solid discretized by a four-node generic shell cell, reducing the integral and considering a finite membrane strain. First, a linear buckling analysis was performed in this paper to calculate the maximum critical load and the buckling mode. The number of integration points for the thickness of the shell was set to 5, and the first 10 eigenvalues were specified to be output in the Buckle analysis step.

In order to ensure that the thermal strain used to replace the intrinsic strain occurs only in the *x*-direction, it is necessary to first obtain the orthotropic thermal expansion coefficients corresponding to different elastic off-axis angles using the strain rotation Equations (7)–(9):(7)$$\varepsilon_{1}^{g}=\frac{\varepsilon_{x}^{g}+\varepsilon_{y}^{g}}{2}+\frac{\varepsilon_{x}^{g}-\varepsilon_{y}^{g}}{2}\cos2\theta -\varepsilon_{xy}^{g}\sin2\theta =K_{1}(T-T_{0})$$
(8)$$\varepsilon_{2}^{g}=\frac{\varepsilon_{x}^{g}+\varepsilon_{y}^{g}}{2}-\frac{\varepsilon_{x}^{g}-\varepsilon_{y}^{g}}{2}\cos2\theta +\varepsilon_{xy}^{g}\sin2\theta =K_{2}(T-T_{0})$$
(9)$$\varepsilon_{12}^{g}=\frac{\varepsilon_{x}^{g}-\varepsilon_{y}^{g}}{2}\sin2\theta +\varepsilon_{xy}^{g}\cos2\theta =K_{12}(T-T_{0})$$
where $\varepsilon_{1}^{g}$, $\varepsilon_{2}^{g}$ are the intrinsic line strains along axis 1 and axis 2 of the principal elastic axis, respectively; $\varepsilon_{12}^{g}$ is half of the intrinsic tangential strain in the coordinate plane of this principal elastic axis; *K*_1_, *K*_2_, and *K*_12_ are the thermal expansion coefficients; and *T* − *T*_0_ is the beginning and end state temperature difference.

When the orthotropic leaf grows in the off-axis angle (*θ* = 0), its principal shear modulus, elastic modulus, and Poisson’s ratio are independent. The variation law of the first three orders of critical intrinsic strain with the principal shear modulus is given below.

The variation in the critical intrinsic strain *β** with the principal shear modulus *G* is shown in Figure 5. It can be observed that the critical instability mode increases with increasing principal shear modulus for different critical instability modes. This is due to the increase in critical strain energy caused by the increase in material stiffness. This increases the critical value of the intrinsic strain required for the transition from planar to curved equilibrium. This is similar to that of thin plates of isotropic materials under unidirectional pressure [23]:(10)$$\sigma_{x}{}^{*}=\frac{k\pi^{2}G}{6(1-\mu ){\left(B/H\right)}^{2}}\;(k=1,2,3\ldots )$$

A law similar to Equation (10) can be obtained from Figure 5, i.e., the critical load is a linear function of the shear modulus for a constant Poisson’s ratio. For isotropic materials, an increase in their shear modulus leads to an increase in Young’s modulus and, thus, does not independently reflect the effect of shear modulus on the critical conditions. In contrast, in orthotropic materials, the principal shear modulus, as an independent material parameter, can be discussed separately with respect to the effect on critical conditions.

### 3.2. Effect of Principal Shear Modulus and Off-Axis Angle on the Post-Buckling Morphology

By studying the critical intrinsic strain *β** of leaf growth in orthotropic materials, it was found that the principal shear modulus *G* plays an important role in the leaf growth instability. With increasing principal shear modulus, it is more difficult for the intrinsic strain to cause instability. Compared with the critical instability of small-deformation linear elasticity, the post-buckling morphology has more complex equilibrium form. Therefore, this section focuses on the influence of the principal shear modulus on the evolution of the leaf post-buckling morphology with increasing intrinsic strain *β*.

Huang et al. [21] found that with increasing intrinsic strain, the post-buckling morphology of isotropic materials is dominated by Saddle Bending, Twisting, Helical Twisting, and Saddle Bending and Edge Waving, in sequence (*n* = 2 and *n* = 10). The post-buckling morphology under the condition of n∈[2, 10] mostly shows the Intermediate State (Saddle Bending and Edge Waving).

The post-buckling morphology shows similar evolution to former isotropic materials during the process of leaf growth in orthotropic material. However, it is different from the isotropic material. The existence of the off-axis angle *θ* makes the leaf in the growth direction present In-Plane Bending and Twisting coupling; under larger *n* conditions, it also has a Twisting and Helical Twisting balance.

A combination of geometric defects of the first 5 buckling modes was introduced in the post-buckling analysis to generate out-of-plane buckling, and the scale of the geometric defects was set to 1% of the thickness to ensure that there was no effect on the post-buckling morphology. In this paper, a small damping factor was set in the post-buckling analysis step to ensure the convergence of the results without affecting the free growth.

The variation in the post-buckling morphology of the material with the principal shear modulus for *n* = 2, 6, and 10 is shown in Figure 6. At the same shear modulus, the intrinsic strain shows a Saddle Bending equilibrium when it exceeds the critical value. This is due to the lower elastic strain energy in Saddle Bending equilibrium compared to the other forms.

At *n* = 2, compared with materials of large shear modulus, the small shear modulus allows the material to transition to the Twisting form as the intrinsic strain increases in order to accommodate the large strain energy associated with the larger intrinsic strain. At this time, the central axis of the material is still straight. As the intrinsic strain continues to increase, the equilibrium of the material in the Twisting form becomes unstable and continues to transition to equilibrium in the Helical Twisting form, because the Helical Twisting form is more adaptable to high intrinsic strain than the Twisting form. As the principal shear modulus increases, the strain energy corresponding to the Twisting form increases. The intrinsic strain required from the Saddle Bending equilibrium to Twisting equilibrium increases. Materials with high shear modulus will maintain a stable Saddle Bending equilibrium in a larger intrinsic strain interval beyond the critical value.

At *n* = 6 and 10, the small-shear-modulus material transforms from the initial Saddle Bending equilibrium to the Intermediate State equilibrium as the intrinsic strain increases. This is because the distribution of the intrinsic strain in the face of the leaf tends to the two long sides of the leaf when *n* is larger. In order to accommodate the increase in elastic strain energy, the long edge of the leaf increases with the intrinsic strain in the form of waving. This transforms the leaf from single Saddle Bending to a combination of Saddle and Wave equilibrium forms, that is, the Intermediate transition form. As the intrinsic strain continues to increase, the long edge of the leaf adopts a continuous undulating Wave form to adapt to the high strain energy from the intrinsic strain. For materials with large shear modulus, it requires higher strain energy from the initial Saddle Bending equilibrium to transition to the complex form. Thus, materials with large shear modulus will maintain a stable Saddle Bending equilibrium for a longer period of time when the intrinsic strain begins to exceed the critical value.

For Figure 7a, the material parameters were set as follows: *E*_1_ = *E*_2_ = 0.35 GPa, *n* = 2. A phase diagram of the variation in the leaf post-bending morphology with the off-axis angle *θ* is given. It can be observed that when the off-axis angle *θ* is close to 0° or 90°, the equilibrium morphology with increasing intrinsic strain *β* transforms from Saddle Bending to Twisting and Helical Twisting. It has little difference from the growth along the elastic principal axis. The leaf transforms more easily to Helical Twisting when the strain increases (as the off-axis angle approaches 22.5°) because the material element has higher tensile–shear coupling at this off-axis angle. As the off-axis angle approaches 45°, the material is predominantly in bending equilibrium. The leaf maintains a stable Saddle Bending equilibrium at higher intrinsic strains.

For Figure 7b, the material parameters were set as follows: *E*_1_ = *E*_2_ = 0.35 GPa, *μ*_12_ = 0.25, *G* = 0.16 GPa, *n* = 6. It can be observed that the Intermediate State, with increasing intrinsic strain, transformed quickly to the Wave morphology in the process of the leaf post-buckling morphology changing with the off-axis angle from 0° to 45°.

## 4. Conclusions

In this paper, the instability of orthotropic leaves due to in-plane non-uniform growth was modeled. The variation in the leaf critical intrinsic strain with the principal shear modulus of the orthotropic material was obtained by finite element analysis. The Young’s modulus and Poisson’s ratio are *E*_1_ = *E*_2_ = 0.35 GPa and *μ*_12_ = 0.25, respectively, for orthotropic materials. Unlike for isotropic materials, the principal shear modulus *G* can be taken independently for orthotropic materials with constant Young’s modulus and Poisson’s ratio. Therefore, compared to that of isotropic materials, the study of the shear modulus of orthotropic material leaves in relation to buckling conditions can more accurately reflect the effect of material stiffness on buckling. The critical intrinsic strain is a linear function with respect to the principal shear modulus under surface power-law exponent conditions *n* = 2 and 10. Similarly, secondary instability occurs less easily with increasing principal shear modulus. The influence of material stiffness on the critical instability condition was revealed. Furthermore, we obtained phase diagrams showing the leaf equilibrium morphology after increasing instability with different principal shear modulus and off-axis angle. They also showed the differences in the evolution of the post-buckling morphology for the leaf with the surface power-law exponent of the intrinsic strain *n*. An interesting phenomenon can be observed in Figure 6 for *n* = 2: When *G* < *E*_1_/2(1 + *μ*_12_), the Saddle Bending and Helical Twisting forms were almost difficult to produce, while as *G* increased to *E*_1_/2(1 + *μ*_12_), the two morphological transition boundary curves changed abruptly, almost simultaneously (*E*_1_ = *E*_2_ = 0.35 Gpa, *μ*_12_ = 0.25, *G* ≈ 0.1 Gpa), and the Twisting interval was quickly compressed. When *G* > *E*_1_/2(1 + *μ*_12_), the transition from the Saddle Bending form to the Helical Twisting form (*E*_1_ = *E*_2_ = 0.35 Gpa, *μ*_12_ = 0.25, *G* ≈ 0.16 Gpa) occurred almost under a large intrinsic strain, which is due to the fact that Twisting is not easily generated under the condition of large torsional stiffness, so that the leaves can maintain stable Saddle Bending. For *n* = 6 or 10, i.e., since the distribution of intrinsic strain in the face is concentrated toward the long edge, large strain energy leads to edge ripples in the material. Comparing *n* = 6 and 10, the Saddle Bending transition to Edge Waving was faster as the surface power-law exponent increased. As with *n* = 2, the transition boundary curve also changed abruptly when *G* ≈ *E*_1_/2(1 + *μ*_12_), and the intrinsic strain required for the Saddle Bending transition to Edge Waving increased rapidly. As *G* continued to increase, the leaf maintained stable Saddle Bending as well. This study shows that soft substances are more likely to wrinkle than hard substances. For off-axis angle *θ* ≠ 0, the phase diagram of the post-buckling morphology also showed a symmetric distribution because the stiffness matrix of the orthotropic material is symmetric about 45°. At this time, the post-buckling morphology evolution is related to both the principal shear modulus and off-axis angle. For off-axis angle *θ* ≈ 22.5°, the material in-plane bending–torsion coupling effect is obvious, so Twisting or Helical Twisting more easily forms. For off-axis angle *θ* = 0°, 45°, or 90°, the material is in a unidirectional stress state, and the leaf can maintain a Saddle under lower intrinsic strain when the bending effect is obvious.

## Figures and Tables

**Figure 1 materials-15-07059-f001:**
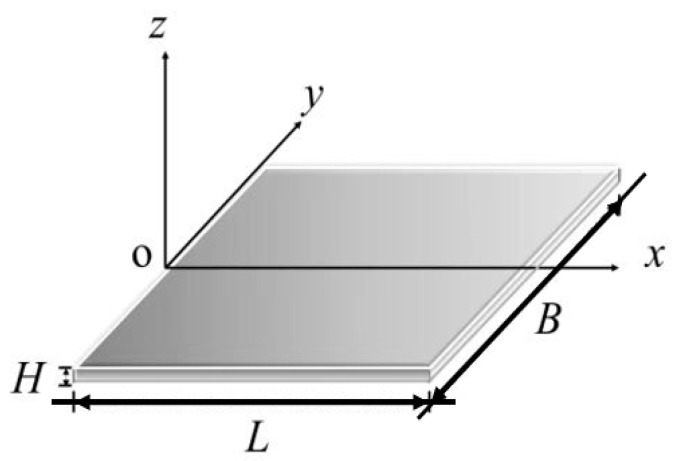
Model of a leaf.

**Figure 2 materials-15-07059-f002:**
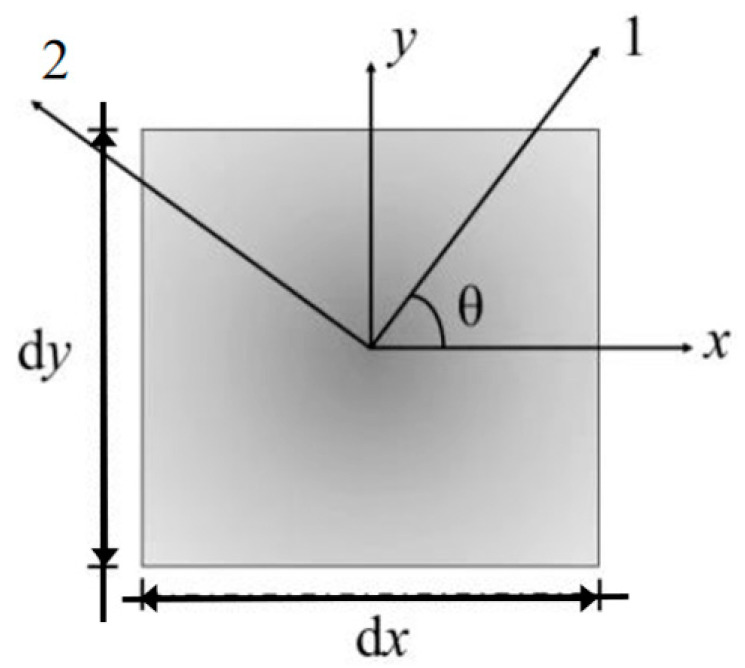
The principal axis directions of an element body in orthotropic material.

**Figure 3 materials-15-07059-f003:**
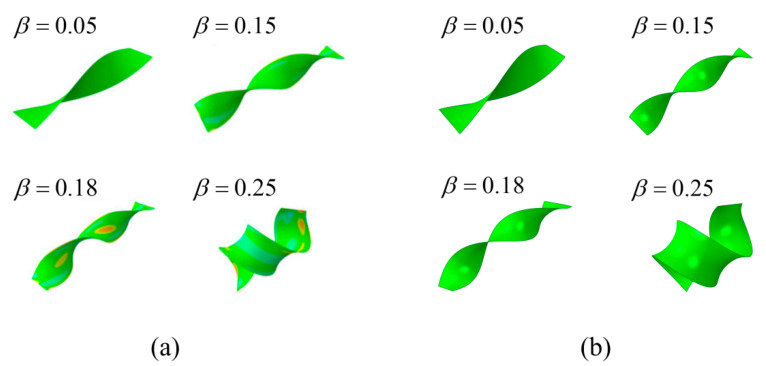
The post-buckling morphology of isotropic material at different intrinsic strains *β* when *n* = 2: (**a**) Huang’s results [21]; (**b**) results in this paper.

**Figure 4 materials-15-07059-f004:**
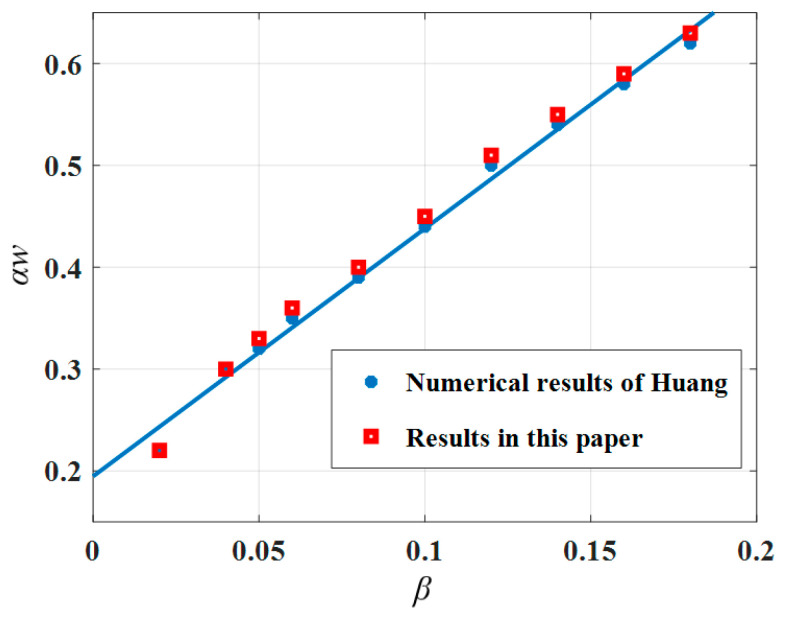
The twist angle per unit length along the stem *αw* with intrinsic strain *β*.

**Figure 5 materials-15-07059-f005:**
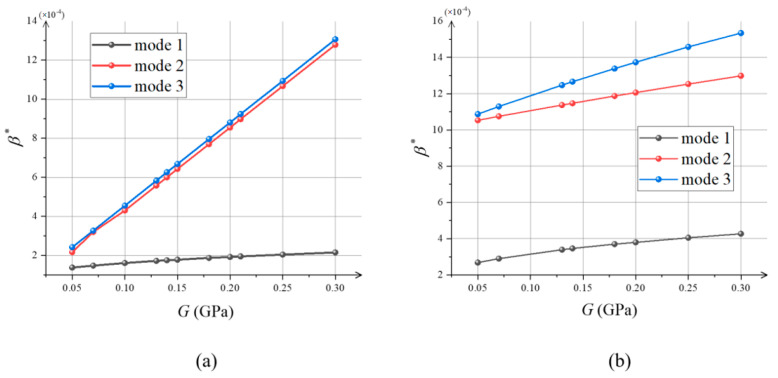
The variation in the critical intrinsic strain *β** of orthotropic materials with the principal shear modulus *G*: (**a**) when *n* = 2; (**b**) when *n* = 10.

**Figure 6 materials-15-07059-f006:**
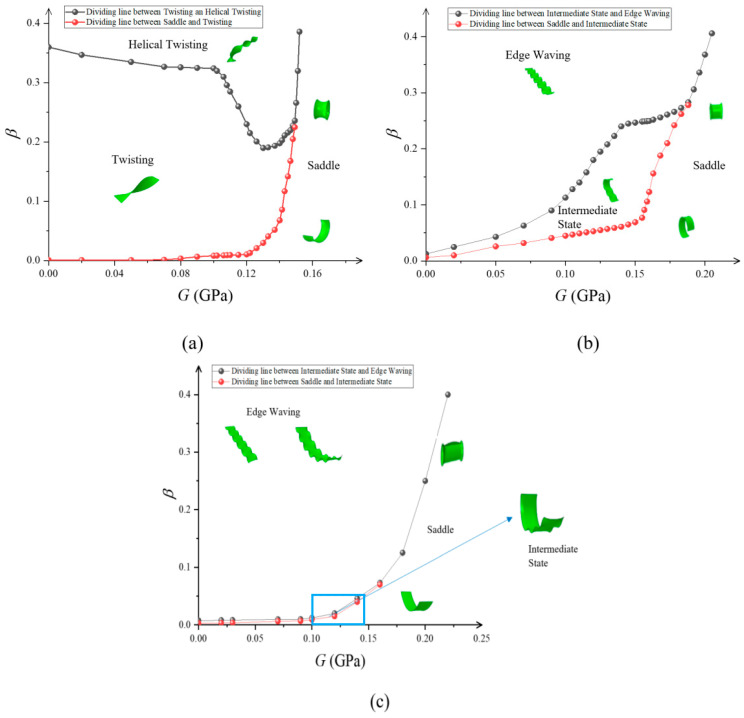
Phase diagrams of variation in the post-buckling morphology of orthotropic materials with intrinsic strain *β* under different principal shear modulus *G*: (**a**) when *n* = 2; (**b**) when *n* = 6; (**c**) when *n* = 10.

**Figure 7 materials-15-07059-f007:**
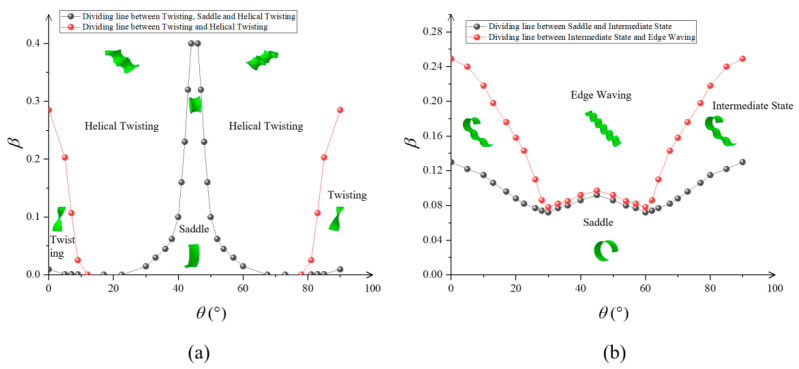
Variation in the rear flexion pattern with off-axis angle *θ*.

## Data Availability

The data used to support the findings of this study are included within the article.

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
