# Peer review of "Study on the Mechanism of Elastic Instability Caused by Natural Growth in Orthotropic Material"

_materials, 2022, doi:10.3390/ma15207059_

Round 1

Reviewer 1 Report

The paper presented by the authors is interesting for the scientific community.

The paper presents 18 bibliographical references, sufficient considering the nature of the research.

The similarity test gave a result of 11%, 2% representing taking over without citing some data from the work of one of the authors.

The value given by the similarity test was acceptable.

I consider that the paper can be published in the form presented.

Author Response

Thank you very much for your support and encouragement.

Reviewer 2 Report

The authors present a study in  the physical mechanisms and man- 10
ifestations of out-of-plane deformation instability  in  naturally grown biological materials. This is a quite interesting topic  with high interest in theliterature. The paper is well presented and discussed. However some points should be improved , before publication.

-the authors should include more details on the numerical procedure  used in the study .  The used softwares are commertial or developed by the authors???

Author Response

Thank you very much for your support and encouragement. In order to reproduce the results better, in this paper, after “4. Conclusions”, a description of the finite element methods (Methods) is added.

Reviewer 3 Report

Dear Authors,

Congratulations for the paper, the contents show hard work. I believe that the only thing missing would be the applications in the industry, to put the progress achieved in context.

Author Response

Thank you very much for your support and encouragement. To make the article perfect, I have added relevant content in Introduction.

Reviewer 4 Report

This is an interesting work; however, prior to proceeding to the next step, the following comments should be addressed by the authors.

1. The language of the manuscript has to be improved.

2. Provide more in-depth discussion of related previous works.

3. Authors should also provide more meaningful discussions regarding the repeatability and reproducibility of the conducted tests/analysis.

4. In the “Conclusion” section, I recommend presenting more quantitative data as the main results of the research study.

Author Response

 Thank you and I am very grateful to your comments for the manuscript. According with your advice, we amended the relevant part in manuscript. Some of your questions were answered below.

1)Q: The language of the manuscript has to be improved

1)A: Thanks for your suggestion, and we have improved the expression of some words.

2)Q: Provide more in-depth discussion of related previous works.

2)A: Thank you for your suggestion, I have added a discussion in 2.3. Model Validation. In unit 3.1, I added a further comparison on studying the buckling of isotropic and orthotropic materials.

3)Q: Authors should also provide more meaningful discussion regarding the repeatability and reproducibility of the conducted tests/analysis.

3)A: First of all, thank you for your suggestion, the results in this paper are based on finite element simulations,and in 4. Conclusions, I added a discussion of the dimensionless results,showing the effect of the principal shear modulus G versus E/2(1+μ) on the postbuckling results. Your comments made us realize that we need to show the similar patterns exist for different material parameters as long as they are all under the above relationship.In order to reproduce the results better, in this paper, after 4. Conclusions, a description of the finite element methods (Methods) is added.

4)Q: In the "Conclusion" section, I recommend presenting more quantitative data as the main results of the research study.

4)A: Thank you for your suggestion, I have added a relevant discussion to this unit.